# CCD Based Detector for Detection of Abrin Toxin Activity

**DOI:** 10.3390/toxins12020120

**Published:** 2020-02-14

**Authors:** Reuven Rasooly, Paula Do, Bradley Hernlem

**Affiliations:** Western Regional Research Center, Foodborne Toxin Detection & Prevention Research Unit, Agricultural Research Service, United States Department of Agriculture, Albany, CA 94710, USA; paula.do@ars.usda.gov (P.D.); bradley.hernlem@ars.usda.gov (B.H.)

**Keywords:** abrin, CCD detector, detection method, Vero cells

## Abstract

Abrin is a highly potent and naturally occurring toxin produced in the seeds of *Abrus precatorius* (Rosary Pea) and is of concern as a potential bioterrorism weapon. There are many rapid and specific assay methods to detect this toxic plant protein, but few are based on detection of toxin activity, critical to discern biologically active toxin that disables ribosomes and thereby inhibits protein synthesis, producing cytotoxic effects in multiple organ systems, from degraded or inactivated toxin which is not a threat. A simple and low-cost CCD detector system was evaluated with colorimetric and fluorometric cell-based assays for abrin activity; in the first instance measuring the abrin suppression of mitochondrial dehydrogenase in Vero cells by the MTT-formazan method and in the second instance measuring the abrin suppression of green fluorescent protein (GFP) expression in transduced Vero and HeLa cells. The limit of detection using the colorimetric assay was 10 pg/mL which was comparable to the fluorometric assay using HeLa cells. However, with GFP transduced Vero cells a hundred-fold improvement in sensitivity was achieved. Results were comparable to those using a more expensive commercial plate reader. Thermal inactivation of abrin was studied in PBS and in milk using the GFP-Vero cell assay. Inactivation at 100 °C for 5 min in both media was complete only at the lowest concentration studied (0.1 ng/mL) while treatment at 63 °C for 30 min was effective in PBS but not milk.

## 1. Introduction

Abrin is a toxin that accumulates in the seeds of the plant *Abrus precatorius* (rosary pea) that grows in a subtropical climate throughout the world. Abrin consists of a heterodimer comprised of a ~30 kDa catalytically active A chain and a ~35 kDa B chain which mediates entry into the cell in a galactose-specific process [1]. The two chains are connected by a single disulfide bond [2]. The 35-kDa chain B is catalytically inactive, but mediates the entry of the A:B holotoxin complex into the cytosol by affinity and attachment to a galactose moiety in the cell membrane [3]. In the cytosol, the Abrin A chain inactivates ribosomes by cleaving the N-glycosidic bond of adenine at nucleotide position 4324 in the 28s rRNA of the 60s ribosomal subunit [4]. This cleavage results in the failure of elongation factor-2 to bind to the ribosome and thus inhibits protein synthesis irreversibly, resulting in cell death. The oral median lethal dose of the toxin ranges from 10 to 1000 µg per kg of body mass when ingested, 3.3 µg per kg when inhaled [5], and 0.7 µg per kg when given intravenously [6].

Several rapid and specific detection methods for abrin have been developed, which include ELISAs [7], immunochromatographic strips [8], multiplex detection immunoassay [9], electrochemiluminescence assays [10], real-time PCR [11], and LC-MS [5]. However, these assays suffer from two main drawbacks. Firstly, they all are unable to distinguish the biologically active and life-threatening form of the toxin from inactivated or degraded forms of the toxin that have lost their ability to impair health. Secondly, they mainly rely upon expensive equipment including thermocyclers, HPLC with mass analysis capabilities, microplate readers, and spectrophotometers. Thus, there is a need for an inexpensive and affordable alternative method to quantitatively measure abrin biological activity. The present study addresses these needs and takes advantage of the fact that biologically active abrin inhibits protein synthesis in mammalian cells and cells that have been transduced with an adenovirus coding reporter gene that can be rapidly and sensitively assayed. The study overcomes the challenges of simple and inexpensive quantitative detection of biologically active abrin toxin by using the reduction of cell fluorescence, colorimetric, or luminescent intensity measured with a low-cost CCD based florescent sensor. This approach enables sensitive detection of biologically active abrin.

## 2. Results

This section may be divided by subheadings. It should provide a concise and precise description of the experimental results, their interpretation, as well as the experimental conclusions that can be drawn.

### 2.1. CCD Based Detector for Abrin Toxin Activity

The major components of the detector are shown in Figure 1. For detection of active abrin toxin, a cell-based assay combined with low-cost, small portable CCD based detector system suitable for multiple optical detection modalities including colorimetric and fluorescent was developed. The field portable battery-operated CCD based detector can read and analyze an entire assay plate at once, converting light into electrical signals. Depending upon the assay, the light can be transmitted with light or fluorescent emission.

### 2.2. Miniature Assay Plates

To accommodate the small field of view (FOV) of the new compact detector, we fabricated miniature assay plates from opaque black polymethylmethacrylate (PMMA) absorbing any stray light to minimize the background signal caused by the stray light that would otherwise exist between the wells of a transparent or translucent plate. Only light directed toward the detector was measured.

In this experiment, we compared two optical detection modalities; fluorescence and colorimetric detection for active abrin. For illumination, we constructed the detector system to employ interchangeable light sources comprised of an LED illuminator that can emit white light or blue light for excitation from 450–495 nm. The excitation spectrum of green fluorescent protein (GFP) contains two peaks; a major peak at 395 nm and another at 475 nm within the blue spectral range of the LED illuminator. The GFP fluorescence spectrum peaks at 509 nm with a shoulder at 540 nm (green light). To increase detection sensitivity and suppress interference from the excitation light a wide band pass excitation filter centered on wavelength at 440 nm and with full bandwidth of ± 40 nm (BG12, Schott), and the second is a narrow band pass filter (D480/30 ×) with center wavelength at 480 nm and full bandwidth of ± 15 nm was placed between the LED illuminator and the test section.

In addition, an emission filter was placed on the end of the lens between the test section and the CCD camera. This emission filter (Intor 535/50/75) has a center wavelength of 535 nm and band width of 50 nm blocking all but the green fluorescent signal from reaching the detector.

For colorimetric detection since the background light from colorimetric and luminescent methods is very low, there is no need to filter out any light. Therefore, when measuring for colorimetric or luminescence we removed the extraneous excitation and emission filter and changed the illumination sources, making this user-friendly portable detector system versatile.

### 2.3. Image Analysis

Digital images of the assay plates were captured by the CCD based detector and processed by the freely available imaging software ImageJ. The numerical value of each pixel in the digital image represents the cells’ fluorescence optical intensity or the intensity of transmitted light at that position in the image.

### 2.4. Cell Based in vitro Detection of Abrin

#### 2.4.1. Colorimetric Measurement for Rapid Quantitative Detection of Abrin

The MTT colorimetric measurement was used to demonstrate proof of principle for rapid screening and quantitative detection of abrin. This assay is based on the fact that abrin toxin suppresses expression of the mitochondrial dehydrogenase enzymes in Vero cells. In the assay system, this dehydrogenase activity reduces the yellow colored water-soluble tetrazolium compound 3-(4,5-dimethylthiazolyl-2)-2,5-diphenyltetrazolium bromide (MTT) to the insoluble compound (E,Z)-(5-(4,5-dimethylthiazol-2-yl)-1,3-diphenylformazan (MTT-formazan) which forms purple colored crystals that cannot pass through the cell membrane and are therefore retained by the cells (Figure 2). MTT has an absorption maximum of 570 nm while MTT-formazan has an absorption maximum of 490 nm.

We constructed a low-cost portable CCD based detector system and fabricated assay plates for this device. The colorimetric results in Figure 3 showed a correlation between abrin concentration and transmission of light passing through the plate well as collected by the CCD detector located above the plate from the LED light source below. The intensity of transmitted light is reported as the average pixel density in each well and this was used to determine the concentration of abrin. As illustrated in Figure 3b, with the increased abrin concentration there is a corresponding increase in the average pixel density as the abrin reduces the production and cellular accumulation of the purple colored formazan product. The increase in transmitted light reaching the detector occurs in a concentration-dependent manner with linear correlation over 3-log range from 1 pg/mL to 1 ng/mL. Specifically, abrin at concentrations of 100, 10, 1, 0.1, 0.01, 0.001 ng/mL resulted in corresponding pixel densities of 250.6 ± 2, 254.9 ± 0.07, 248.6 ± 4, 235.3 ± 4, 198 ± 9, and 167.1 ± 2, respectively. The limit of detection of this colorimetric assay for abrin is 10 pg/mL which is 10^6^ times lower than the LD50 value of 10 µg/kg when ingested.

We further confirmed the low-cost device results (Figure 4a) with a commercial spectrophotometric plate reader with a filter for optical density at 540 nm (OD 540) to measure abrin’s effect on dehydrogenase cell activity (Figure 4b). Since optical density is inversely related to light transmission, plate reader results reported in Figure 4b show a negative relationship between OD 540 and abrin concentration. A t-test analysis shows that the limit of detection of abrin by a commercial plate reader was 0.1 pg/mL compared to without abrin, which is 100 times more sensitive than the CCD detector which can detect 10 pg/mL of biologically active abrin.

#### 2.4.2. Relationship between Data Acquired by a Commercial Plate Reader and Low-cost CCD Detector

Figure 5 reports the correlation between data from a commercial plate reader measuring optical density plotted on the x-axis and data from the low-cost CCD detector measuring inverted average pixel density plotted on the y-axis for serial dilutions of abrin toxin. The coefficient of determination with R-square of 0.95 meant that there is 95% agreement between the commercial plate reader and low-cost CCD detector. This suggests that low-cost CCD detector is an effective technology to the commercial plate reader for detection of biologically active abrin.

#### 2.4.3. Fluorescence Measurement for Rapid Quantitative Detection of Abrin

In search for a method that can increase the assay sensitivity, transduced Vero and HeLa cell lines that were used for fluorescence detection was compared between the commercial fluorometric plate reader employing a photomultiplier tube and the CCD detector system. These modifications of the colorimetric assay are based on the interference of abrin with the GFP protein synthesized in transduced Vero and HeLa cell lines that express the GFP reporter gene. The effect of abrin exposure results in a quantitative decrease in GFP fluorescence. The relatively expensive commercial fluorometer plate reader contains a photomultiplier which comprises a photosensitive surface producing a stream of photoelectrons in response to photons, generating an amplified electric signal. Such photomultipliers are spot sensors that are confined to respond to light from a tiny area, quantifying fluorescence from only one sample well at a time. Consequently, a plate must be scanned by constantly moving either the detector, optics, or sample plate so all the wells can be read by the spot detector. The assay fluorescence data using the plate reader are shown in Figure 6a for Vero cells and Figure 6b for HeLa cells and are presented in relative fluorescence units (RFU) showing that the GFP fluorescence emission intensity decreases in both cell lines in a concentration dependent manner. Therefore, both cells lines are suitable for quantification of abrin. However, the level of GFP expression in the Vero cell line was seven times higher than in HeLa cell line. The results also show that Vero cells had a lower detection limit for abrin than HeLa cells. Incubation of Vero cells (**a**) with abrin at the low concentration of 0.1 pg/mL induced a significant decrease (P<0.05) in GFP fluorescence emission intensity from control. However, in HeLa cells there was no significant difference at 0.1 pg/mL, however, there was a significant difference at 100 pg/mL (**b**). Specifically, Vero cells exposed to decreasing abrin concentrations of 1, 0.1, 0.01, 0.001, 0.0001 and 0 ng/mL resulted in increasing in fluorescence intensity measurements corresponding RFUs of 2638.7 ± 40.3, 2922.0 ± 34.9, 3697.67 ± 145.3, 6928.7 ± 302.8, 12061.7 ± 381.2, and 15928.7 ± 556.4, respectively. In HeLa cells the result show inverse sigmoid dose-response curve. Resulting RFUs were 1470.3 ±11.3, 1886.3 ± 27.3, 2236.3 ± 139.0, 2273.0 ± 42.3, 2371.7 ± 64.6, and 2163.3 ± 64.7 for the same concentrations of abrin, respectively. This result suggest that the sensitivity of this assay might be increased by using a cell line that can express higher level of GFP and respond to abrin at lower concentrations.

#### 2.4.4. Quantitative Fluorescence Digital Imaging Analysis of Abrin

To demonstrate the potential of low-cost CCD detectors for abrin biological activity detection we illuminated the plate with blue light for excitation and the wells were imaged by the CCD detector as shown in Figure 7a. The average pixel density (average brightness) from the digital images of three replicates representing the optical intensity of cellular fluorescence emission was analyzed from a series of image frames using the freely available imaging software ImageJ and was plotted against various abrin concentrations. The results in Figure 7b,c show a correlation between an increase in abrin concentration resulting in a decrease in fluorescence intensity emission. Incubation of Vero cells with abrin at the low concentration of 100 pg/mL induced a significant decrease (*p* < 0.05) in GFP fluorescence emission intensity from control. Specifically, transduced Vero cells exposed to abrin at concentrations of 1, 0.1, 0.01, 0.001, 0.0001, and 0 ng/mL resulted in the corresponding average pixel density of 6.1 ± 0.8, 25.4 ± 5.5, 70.9 ±13.7, 73.1 ± 1.9410, and 80.8 ± 10.4, respectively.

#### 2.4.5. Correlation between CCD Detector and Commercial Fluorometer Using Photomultipliers

In order to measure the strength of a linear relationship between the CCD detector and fluorometer measurements the normalized average signal brightness that was detected by a charge-coupled device (CCD) and analyzed with ImageJ was plotted against the normalized RFU values of the fluorometer (Figure 8). These results show that an increase in average pixel density values corresponds to an increase in RFU and this positive correlation between CCD measurements and fluorometer has R-squared value of 0.79 (R^2^ = 0.79). These results demonstrate that the low-cost CCD detector is an effective technology for detecting abrin activity at comparable levels to a more expensive commercial fluorometer.

#### 2.4.6. Thermal Inactivation of Abrin

Pasteurization is a common food processing operation used to inactivate pathogens, spoilage organisms, and heat labile toxins. We examined the effects of several heat treatments on the activity of abrin in PBS and milk. Abrin concentrations ranged from 0.1 to 100 ng/mL and controls with no toxin present were also included. Samples were either untreated (unheated) or heat treated at 63 °C for 30 min, 72 °C for 15 s, 89 °C for 1 s, or 100 °C for 5 min. Following treatment, the samples were assayed for toxin activity using transduced Vero cells. The results are presented in Figure 9 and show that pasteurization has limited effect on abrin activity and then only at the lowest concentrations and in milk only when treated for 5 min at 100 °C. In both PBS and milk, abrin at a concentration of 0.1 ng/mL was fully inactivated under those conditions. Likewise, abrin at the same concentration in PBS was fully inactivated when subjected to pasteurization at 63 °C for 30 min.

## 3. Discussion

This work is novel in several ways, specifically that a low-cost CCD detector system is utilized in conjunction with colorimetric and fluorescent measurements for the quantitative detection of biologically active abrin. Abrin quantitatively suppresses cellular mitochondrial respiration which can be measured using the colorimetric MTT-formazan assay. Abrin also inhibits the expression of GFP in transduced Vero and HeLa cells, which fluorescence can be quantified as a measure of abrin activity. The portable and low-cost CCD detector system described here is relatively simple and costs 30 times less than a plate reader, can detect a large number of samples simultaneously, and has a limit of detection of 10 pg/mL which is 10^6^ times lower than the LD50 value by ingestion (10 µg/kg). With a limit of detection of 10 pg/mL, the method presented here is a 200-fold improvement in sensitivity over the recently developed quantitative high-resolution targeted mass spectrometry method reported to have a limit of detection of 2 ng/mL [12]. Further, this relatively new MS-based assay is not easily scalable and relies upon very expensive equipment and skilled personnel, therefore, it can only be performed in very few laboratories. However, most importantly it fails in the vitally important feature of the method presented here; it cannot distinguish between the biologically active and life-threatening form of abrin from inactivated abrin which has lost its ability to impair health.

Previous activity studies using a mouse bioassay have found that abrin is highly resistant to thermal inactivation and indicate that some commonly used pasteurization methods for milk have limited effect on abrin inactivation [13]. Our studies support these claims and show that milk exhibits a protective effect on abrin. Treatment at 63 °C for 30 min was not effective in milk but inactivated abrin in PBS. However, when we applied higher thermal treatment of 100 °C and at lower abrin concentration of 0.1 ng/mL milk lost its protective effect and abrin was fully inactivated. Adopting this portable, relatively simple, low-cost CCD detector system may allow expanded testing for biologically active abrin to promote food safety especially where resources are limited.

## 4. Materials and Methods

### 4.1. CCD Based Detector system

The CCD based detector system consists of (1) Point Grey Research *Chameleon* monochrome camera equipped with a C-mount CCTV lens used as the photodetector used to measure GFP fluorescence or transmitted light, (2) Pentax 12 mm f1.2 lens, item # C61215KP (Spytown, Utopia, NY, USA), (3) green emission filter HQ535/50 m (Chroma Technology Corp Rockingham, VT), (4) blue excitation filter D486/20x (Chroma Technology Corp, Rockingham, VT), and (5) LED illumination box containing blue and white LEDs custom built by Luminousfilm (Shreveport,Louisiana, www.luminousfilm.com/led.htm). Excitation and emission filters did not use colorimetric measurement of transmitted light.

The plate assay material consists of (1) black 3.2 mm acrylic (Piedmont Plastics, Beltsville, MD), (2) clear 0.5 mm polycarbonate film (Piedmont Plastics, Beltsville, MD), (3) 3 M 9770 adhesive transfer tape (Piedmont Plastics, Beltsville, MD), and (4) Epilog Legend CO2 65 W cutter (Epilog, Golden, CO).

#### Computer Control and Data Analysis

Images were captured using PC (laptop or desktop) with USB port, and analyzed by ImageJ analysis software source NIH software (U.S. National Institutes of Health, Bethesda, MD, USA,). (http://rsb.info.nih.gov/ij/download.html). The data were transferred to Excel (Microsoft, 2012, Redmond, WA, USA) and SigmaPlot (version 10, Ashburn, VA, USA) data analysis software.

#### Biological and Cell Culture Reagents

Abrin was obtained from Toxin Technology (Sarasota, FL, USA). Vero cells: African Green Monkey adult kidney cells (ATCC CCL-81), Homo sapiens HeLa cells (ATCC CCL-2), and HEK293 cells (ATCC CRL-1573) were obtained from American Type Culture Collection (Manassas, VA, USA). Dulbecco’s modified eagle’s medium (DMEM) were obtained from Gibco (Carlsbad, CA, USA), fetal bovine serum (FBS) and fetal calf serum (FCS) were obtained from Hyclone (Waltham, MA, USA). 

#### Colorimetric Measurement Reagents

MTT (3-(4,5-dimethylthiazol-2-yl)-2,5-diphenyltetrazolium bromide) and Dimethyl sulfoxide (DMSO) were obtained from Sigma-Aldrich (St. Louis, MO, USA)

### 4.2. Creation of Adenoviral Vectors that Express the Green Fluorescent Protein (GFP) Gene

To show abrin toxin effect on Vero cells we quantified changes in GFP expression levels. The 750 bp fragment of the GFP gene that was isolated from the Green Lantern vector (Life technologies, Carlsbard, CA, USA) was subcloned into an adenoviral shuttle plasmid between the cytomegalovirus immediate early promoter and the polyadenylation signal from bovine growth hormone. Since HEK293 cells express the E1 regen gene of adenoviral, they can be used to propagate adenoviral vectors in which the E1 gene is deleted. To create the adenoviral vectors the plasmid pJM17 containing the full length of the adenovirus 5 genome plus a 4.4 kb sequence of antibiotics resistance gene were cotransfected in HEK293 cells with the GFP shuttle plasmid that part of the E1 region substituted by the GFP gene. After 10 days, due to cytopathic effect the transfected cells became round and detached from the plate. Both cell pellet and supernatant were collected. The presence of the adenovirus expressing the GFP gene were examined by fluorescence microscopy.

### 4.3. Colorimetric Assay of Abrin Activity

Untransduced Vero cells were introduced to a test plate at a cell density of 1 × 10^4^ cells per well in 100 μL of medium. The cells were allowed to attach to the plate during an overnight incubation, after which time 5 μL aliquots of a range of concentration of abrin were added to each well. The cells were incubated for a further 18 h in an incubator held at 37 °C with an atmosphere of 5% CO_2_ in air. Cells were washed twice in PBS and resuspended in 100 μL of PBS. The plate reader was used to measure optical density and low-cost CCD detector system was used to measure average pixel density. A 2 mg/mL MTT solution was prepared in PBS. Aliquots of 25 μL MTT solution were added to each well, with nontransduced Vero cells, followed by a 4 h incubation at 37 °C after which time the medium was removed from the adherent cells. Aliquots of 100 μL of dimethyl sulfoxide (DMSO) were added to each well to solubilize the purple colored product and this was then analyzed by plates read at 540 nm or by the low-cost CCD detector system.

### 4.4. Fluorometric Assay of Abrin Suppression of GFP

Transduced Vero or HeLa cells were introduced to a test plate at a cell density of 1 × 10^4^ cells per well in a total volume of 100 μL. To 95 μL of medium and cells in each well, 5 μL aliquots of a range of concentration of abrin or test sample were added. The cells were incubated for 18 h in an incubator held at 37 °C with an atmosphere of 5% CO_2_ in air. Cells were washed twice in PBS and resuspended in 100 μL of PBS and reintroduced to a test plate. Fluorescence of GFP was measured by the CCD detector system or by fluorometer plate reader.

## Figures and Tables

**Figure 1 toxins-12-00120-f001:**
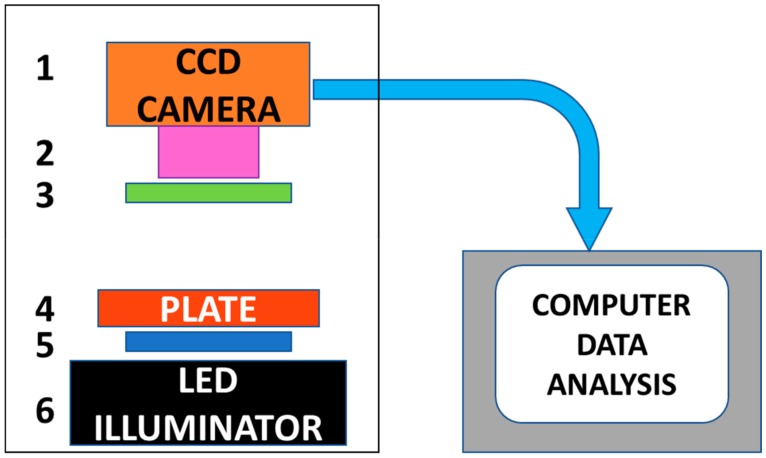
Schematic of CCD based detector: 1. CCD camera, 2. 12 mm wide angle lens, 3. excitation filter (for fluorescence detection), 4. assay plate, 5. emission filter (for fluorescence detection) 6. LED illuminator with white, blue, red, and green LEDs.

**Figure 2 toxins-12-00120-f002:**
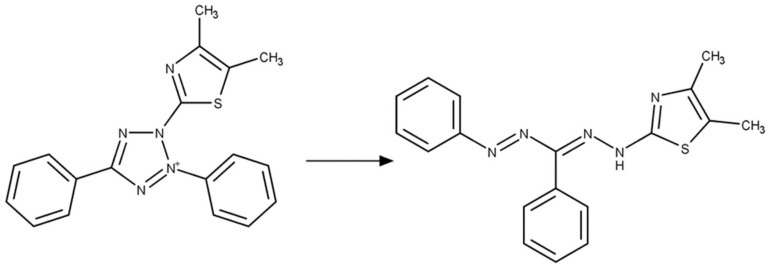
Mitochondrial dehydrogenase catalyzed reduction of yellow colored MTT (left) into the purple colored MTT-formazan product (right).

**Figure 3 toxins-12-00120-f003:**
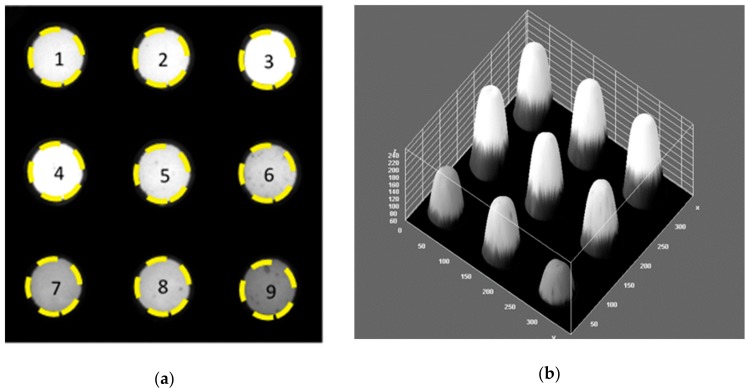
Colorimetric detection of abrin. (**a**) An image of the laminated assay plate containing increasing abrin concentrations. The control (0 ng/mL) is well #9 and tenfold increasing concentrations of abrin are in wells #8 (0.1 pg/mL) through well #1 (100 ng/mL). The intensity of transmitted light was detected by the low-cost CCD detector (**b**) the corresponding quantified image analysis.

**Figure 4 toxins-12-00120-f004:**
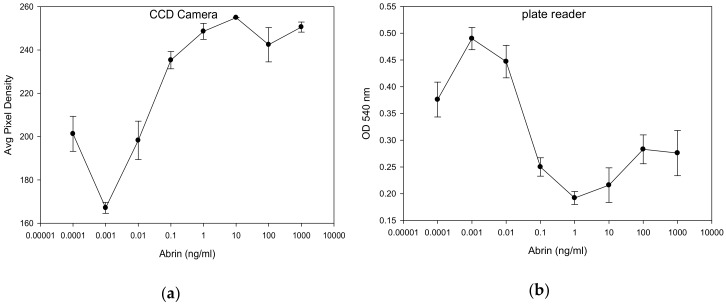
Quantitative determination of the biological activity of abrin by accumulation of the purple colored formazan product. Vero cells were incubated for 24 h with various concentrations of abrin and then treated with MTT. The light transmittance through formazan was detected by the low-cost CCD detector measured in units of average pixel density (**a**). The light absorbed by formazan was detected by a plate reader (**b**) measured as optical density (OD) at 540 nm. Error bars represent standard errors.

**Figure 5 toxins-12-00120-f005:**
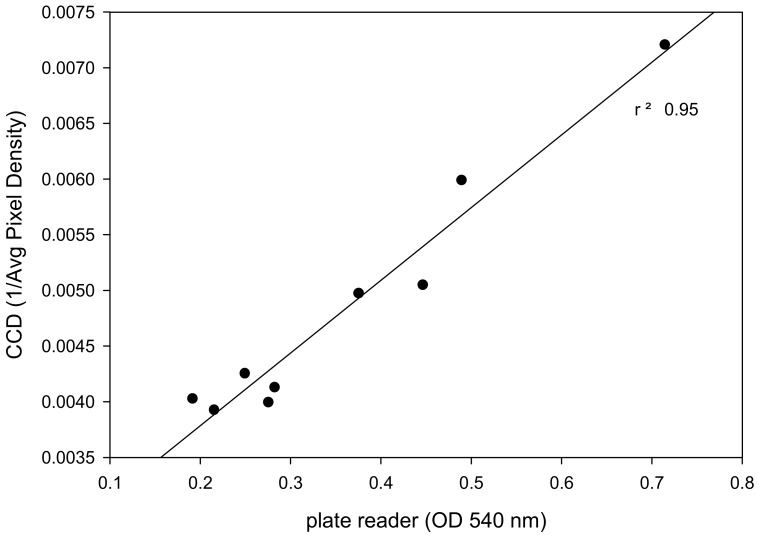
Correlation between commercial plate reader (OD 540 nm) and CCD detector (1/Average pixel density) to serial dilutions of abrin toxin.

**Figure 6 toxins-12-00120-f006:**
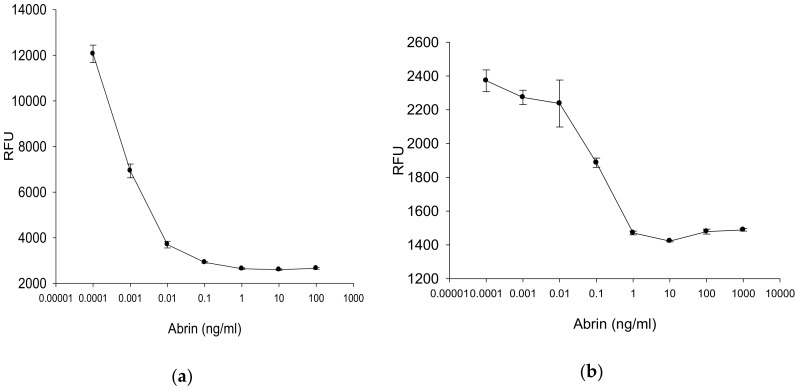
Transduced Vero (**a**) and HeLa (**b**) cell lines were treated with various concentrations of abrin toxin for 24 h. The fluorescence signals from the plate were detected by the commercial plate reader. Error bars represent standard errors.

**Figure 7 toxins-12-00120-f007:**
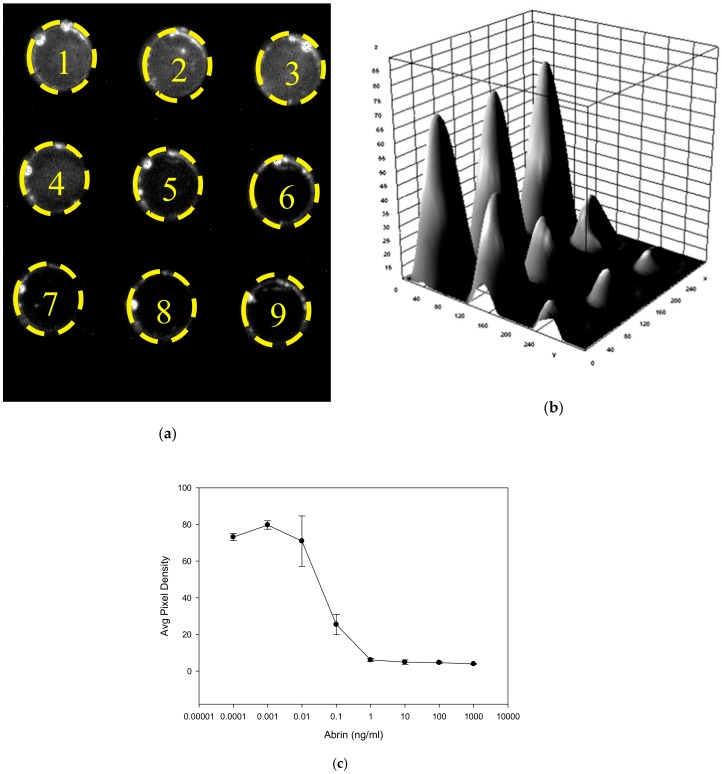
Fluorescence detection of abrin. (**a**) An image of the laminated assay plate containing increasing abrin concentrations. The control (0 ng/mL) is well #9 and tenfold increasing concentrations of abrin are in wells #8 (0.1 pg/mL) through well #1(100 ng/mL). The intensity of transmitted light was detected by the low-cost CCD detector (**b**) the corresponding quantified image analysis, and (**c**) the plotted data.

**Figure 8 toxins-12-00120-f008:**
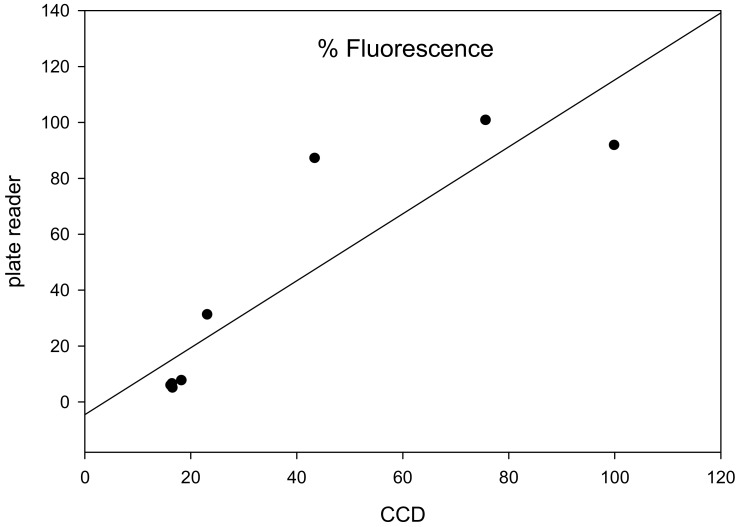
Correlation between CCD based detection and commercial fluorometer measurement. After incubation of transduced Vero cells for 24 h with serial dilutions of abrin toxin, the fluorescence intensity (brightness) measured by the CCD detector and reported as average pixel density was normalized and plotted against the normalized signal measured by the fluorometer as reported as RFU.

**Figure 9 toxins-12-00120-f009:**
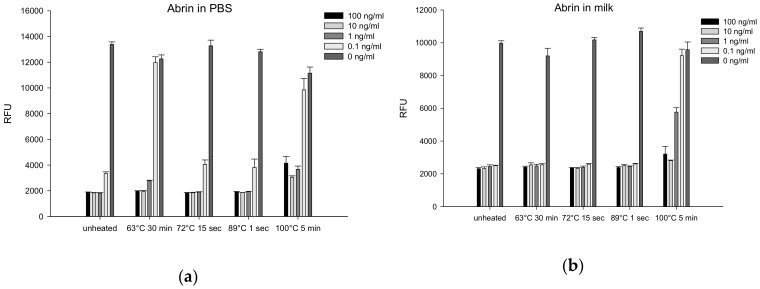
Effect of pasteurization on the biological activity of abrin toxin. PBS (**a**) and milk (**b**) samples were spiked with abrin toxin for final concentrations of 0 ng/mL or from 0.1 to 100 ng/mL and pasteurized at 63 °C for 30 min, 72 °C for 15 s, 89 °C for 1 s, 100 °C for 5 min, or unheated. Five microliters of treated sample with 95 μL of media were incubated with Vero cells for 18 h and subsequently inhibition of GFP expression was quantified fluorometrically. Error bars represent standard errors, and an asterisk (*) indicates significant differences (*p* < 0.05) between spiked and unspiked milk.

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
