# Peer review of "CCD Based Detector for Detection of Abrin Toxin Activity"

_toxins, 2020, doi:10.3390/toxins12020120_

Round 1

Reviewer 1 Report

In the manuscript authors describe a simple and low-cost CCD detector system for detection of abrin toxin activity. Authors evaluate the method with colorimetric and fluorometric cell-based assays for abrin activity, namely by measuring the abrin suppression of mitochondrial dehydrogenase and abrin suppression of green fluorescent protein (GFP) expression in transduced Vero and HeLa cells. The main advantage of the presented system are cells that have been transduced with an adenovirus coding reporter gene and thereby do not require transfection with reporter mRNA.

Although authors provide interesting and novel data, I have several major objections:

Since translation inhibition is a downstream effect of depurination, translation inhibition assays do not measure catalytic activity of the RIPs on the ribosome directly. Therefore, to validate properly the presented alternative method of quantitative measurement of abrin biological activity, I would like to see additionally one of the depurination assays, ex. qRT-PCR assay, to detect the catalytic activity of RIPs under physiological conditions using ribosomes as substrates.

What is more, recently published paper about the influence of ricin-mediated rRNA depurination on the translational machinery in vivo says: “depurination does not affect notably any particular step of translation, and translational slowdown caused by ricin is not a direct consequence of depurination and cannot be considered as the sole source of cell death.  Instead, SRL depurination in a small fraction of ribosomes blocks cell cycle progression with no effect on cell viability.” – How authors comment their results on that?

Secondly, in the final “results and discussion” section, I would love to see more examples and arguments for the need of inexpensive and alternative method to quantitatively measure abrin biological activity in the field.

Author Response

Dear Damia Dou Assistant Editor,

Thank you for considering our manuscript title: “CCD based detector for detection of Abrin toxin activity” (ID: toxins-706881) for publication as a research article in Toxins.  We have addressed the questions posed by the Reviewers.

In the manuscript authors describe a simple and low-cost CCD detector system for detection of abrin toxin activity. Authors evaluate the method with colorimetric and fluorometric cell-based assays for abrin activity, namely by measuring the abrin suppression of mitochondrial dehydrogenase and abrin suppression of green fluorescent protein (GFP) expression in transduced Vero and HeLa cells. The main advantage of the presented system are cells that have been transduced with an adenovirus coding reporter gene and thereby do not require transfection with reporter mRNA.

Although authors provide interesting and novel data, I have several major objections:

Since translation inhibition is a downstream effect of depurination, translation inhibition assays do not measure catalytic activity of the RIPs on the ribosome directly. Therefore, to validate properly the presented alternative method of quantitative measurement of abrin biological activity, I would like to see additionally one of the depurination assays, ex. qRT-PCR assay, to detect the catalytic activity of RIPs under physiological conditions using ribosomes as substrates.

Our aim in this paper was only to describe a simple and low-cost CCD detector system for detection of abrin toxin activity and not to question or elucidate the mechanism of abrin’s activity that we show leads to the suppression of mitochondrial dehydrogenase activity and suppression of green fluorescent protein (GFP) expression. It is sufficient for the purposes of the assay for quantification of abrin toxin activity. Therefore, we do not find a need in the context of this manuscript to explore depurination assays.  

This work has already been reported in the case of the similar toxin, ricin. See Szajwaj, M., et al., BBA Molecular Cell Research 1866 (2019) 118554.

What is more, recently published paper about the influence of ricin-mediated rRNA depurination on the translational machinery in vivo says: “depurination does not affect notably any particular step of translation, and translational slowdown caused by ricin is not a direct consequence of depurination and cannot be considered as the sole source of cell death.  Instead, SRL depurination in a small fraction of ribosomes blocks cell cycle progression with no effect on cell viability.” – How authors comment their results on that?

It is beyond the scope of the current study to elucidate the exact mechanism of abrin’s activity. Endo et al., 1987 reported that abrin cleaves the N-glycosidic bond of adenine at nucleotide position 4324 in the 28s rRNA of the 60s ribosomal subunit, affecting translation and inhibiting protein synthesis, see reference number 4. We added the recently published paper as a new reference reporting that depurination by ricin does not affect notably any particular step of translation and translational slowdown is not a direct consequence of depurination. The authors of that work assert that this mechanism cannot be considered the sole source of cell death.  To the extent that abrin acts by similar mechanisms to ricin, this observation may apply in this case, as well.

Secondly, in the final “results and discussion” section, I would love to see more examples and arguments for the need of inexpensive and alternative method to quantitatively measure abrin biological activity in the field.

The data presented here show that a low-cost CCD camera that cost 30 times less than a plate reader in combination with Vero or HeLa cells can be applied for quantitative detection of biologically active abrin. This low-cost and relatively simple assay that can detect large number of samples simultaneously has a limit of detection 10 pg/mL which is 200 times more sensitive than MS-based assay (Hansbauer et al., 2017) which cannot discern active from inactive toxin and relie upon very expensive equipment and skilled personnel. Thus, adopting this low-cost method allows for expanded abrin testing for the promotion of food safety especially where resources are limited.

Best regards,

Reuven Rasooly

Reviewer 2 Report

The manuscript describes the development of an alternative, CCD based detector for the detection of Abrin toxin. The manuscript could be significantly improved by comparing their technique with other existing techniques (e.g. sensitivity, selectivity, time to get the results, portability, application to other toxin isoforms or closely related Abrus precatorius agglutinin…) (e.g. Hansbauer, 2017; Suzanne, 2019) in the discussion and providing additional background information in the introduction (only 11 references cited in this paper). Specific comments are listed as below:

1. Figure 4a. Why the pixel density measured at 0.1 pg/ml of Abrin was higher than that at 1 pg/ml?

2. I only see a linear correlation of average pixel density vs abrin at concentration range from 1 pg/ml to 1 ng/ml. The intensity is approaching plateau at higher concentration. Correct sentence 160–161.

3. Is the value reported in Figure 4a corrected from pixel density measured at 0 ng/ml of abrin?

4. Figure 7 and line 229–232: A figure showing fluorescence intensity vs abrin conc. should be included.

5. Paragraph/paragraphs of conclusion/summary/discussion should be added following the description of the results.

6. Is this assay selective to detect abrin or is it sensitive to other toxins or chemicals (e.g. ricin)? Please discuss.

Minor comments:

1. Correct the label PLATE in figure 1.

2. Line 22: Abrin consists of a single heterodimer, not two heterodimers.

3. Missing label in figure 6, left panel: Vera-GFP cells with Abrin toxin

Author Response

Thank you for considering our manuscript title: “CCD based detector for detection of Abrin toxin activity” (ID: toxins-706881) for publication as a research article in Toxins.  We have addressed the questions posed by the Reviewers.

Top of Form

The manuscript describes the development of an alternative, CCD based detector for the detection of Abrin toxin. The manuscript could be significantly improved by comparing their technique with other existing techniques (e.g. sensitivity, selectivity, time to get the results, portability, application to other toxin isoforms or closely related Abrus precatorius agglutinin…) (e.g. Hansbauer, 2017; Suzanne, 2019).

Hansbauer et al., 2017 develop quantitative high-resolution targeted mass spectrometry methods for abrin detection with a limit of detection of 2 ng/mL. However, this new MS-based assay relies upon expensive equipment and is unable to distinguish between the biologically active and life-threatening form of the abrin from inactivated abrin that has lost its ability to impair health. Further, this method has a limit of detection of 2 ng/mL whereas our affordable alternative colorimetric assay that utilizes a low-cost CCD detector system has a limit of detection 10 pg/mL. The latter is a 200 times improvement in sensitivity over the mass spectrometry method with the vital benefit of being able to distinguish between biologically active abrin and inactive toxin that has lost its ability to impair health.

in the discussion and providing additional background information in the introduction (only 11 references cited in this paper). Specific comments are listed as below:

Figure 4a. Why the pixel density measured at 1 pg/ml of Abrin was higher than that at 1 pg/ml?

It is unknown why there appears to be a larger effect at 0.1 pg/mL than at 1 pg/mL, however, both CCD device and plate reader show the same result albeit presented in the figure in inverse units; i.e. transmitted light versus absorbed light. It is apparent that the dose-dependent range is poorly extrapolated at both high and low concentrations of abrin.

I only see a linear correlation of average pixel density vs abrin at concentration range from 1 pg/ml to 1 ng/ml. The intensity is approaching plateau at higher concentration. Correct sentence 160–161.

As suggested by the reviewer we have amended this sentence to indicate the linear range over 3-log from 1 pg/ml to 1 ng/ml.

Is the value reported in Figure 4a corrected from pixel density measured at 0 ng/ml of abrin?

The control (0 ng/mL) pixel density was not subtracted from the other data. The data were presented with no modification or adjustment to the control value.

Figure 7 and line 229–232: A figure showing fluorescence intensity vs abrin conc. should be included.

We have created a plot of the data and included it as a figure 7c.

Paragraph/paragraphs of conclusion/summary/discussion should be added following the description of the results.

As requested, we had added a Discussion separate from the Results.

Is this assay selective to detect abrin or is it sensitive to other toxins or chemicals (e.g. ricin)? Please discuss.

We expect that this assay is sensitive to other toxins that inhibit protein synthesis such as ricin, aflatoxin b1 and Shiga toxin. We have discussed this in the Discussion. This assay can be combined with a non-activity assay, such as ELISA, that can confirm the presence of the toxin and this assay can demonstrate whether the toxin is active or inactivated. This would be particularly useful in the case of research into food processing methods to inactivate toxin.

Minor comments:

Correct the label PLATE in figure 1.

Figure includes label “PLATE”.

Line 22: Abrin consists of a single heterodimer, not two heterodimers.

As requested, we have edited and corrected this error.

Missing label in figure 6, left panel: Vera-GFP cells with Abrin toxin

The figure has been edited so that the cell type is specified in the legend rather than in the figure.

Best regards,

Round 2

Reviewer 2 Report

All my comments and suggestions were addressed in a satisfactory way.